# Room-temperature X-ray fragment screening with serial crystallography

Sebastian Günther [1] ✉, Pontus Fischer [1], Marina Galchenkova[1], Sven Falke [1], Patrick Y. A. Reinke [1], Sreevidya Thekku Veedu [1], Ana Carolina Rodrigues [1], Johanna Senst [1], Daniel Elinjikkal [1], Lars Gumprecht[1], Jan Meyer [1], Henry N. Chapman [1,2,3], Miriam Barthelmess[1] & Alke Meents [1] ✉

Structural insights into protein-ligand interactions are essential for advancing drug development, with macromolecular X-ray crystallography being a cornerstone technique. Commonly X-ray data collection is conducted at cryogenic temperatures to mitigate radiation damage effects. However, this can introduce artifacts not only in the protein conformation but also in protein-ligand interactions. Recent studies highlight the advantages of room-temperature (RT) crystallography in capturing relevant states much closer to physiological temperatures. We have advanced fixed-target serial crystallography to enable high-throughput fragment screening at RT. Here we systematically compare RT fragment screening of the Fosfomycin-resistance protein A from *Klebsiella pneumoniae* (FosAKP), an enzyme involved in antibiotic resistance, with conventional single crystal data collection at cryogenic temperature (cryo). With RT serial crystallography we achieve resolutions comparable to cryogenic methods and identify a previously unobserved conformational state of the active site, offering additional starting points for drug design. For ligands identified in both screens, temperature does not have an influence on the binding mode of the ligand. But overall, we observe more binders at cryo, both at physiologically relevant and non-relevant sites. With the potential for further automation, RT screening with serial crystallography can advance drug development pipelines by making undiscovered conformations of proteins accessible.

Structure based drug discovery is a powerful technique for drug development[1,2]. In contrast to biological assay-based screening techniques, in structure-based drug development potential binders are designed based on the 3-dimensional atomic structure of a target protein. Whereas it has recently become possible to predict the 3-dimensional structure of proteins with high reliability and the prediction of binding interactions between a protein target and smaller chemical compounds such as drugs is also advancing[3], it remains challenging and still relies on experimental validation.

Among other techniques, macromolecular X-ray crystallography (MX) using synchrotron radiation provides unmatched throughput in combination with near-atomic resolution in 3D for studying protein-ligand interactions[4,5]. Due to its high demand, MX is highly standardized and structure determination can be performed typically within a

[1]Center for Free-Electron Laser Science CFEL, Deutsches Elektronen-Synchrotron DESY, Notkestr. 85, Hamburg, Germany. [2]The Hamburg Centre for Ultrafast Imaging, Luruper Chaussee 149, Hamburg, Germany. [3]Department of Physics, University of Hamburg, Luruper Chaussee 149, Hamburg, Germany. ✉e-mail: sebastian.guenther@desy.de; alke.meents@desy.de

few minutes if a crystal of sufficient size and quality is available[4]. An alternative technique for structure determination that is becoming more popular is cryo-electron microscopy of single (that is, uncrystallized) macromolecules.

Until today, the majority of structure determinations via MX and all structure determinations via cryo-electron microscopy have been performed at cryogenic temperatures (cryo)[6]. As a result, our current picture about protein structures, structural biology in general, and our derived understanding of enzyme reactions, refer to these temperatures.

In contrast to experiments at cryo, X-ray screening experiments at room-temperature (RT) have the strong advantage that they provide structural information much closer to physiological temperatures. Therefore, binding interactions between ligands and target proteins observed at RT are expected to be physiologically more relevant and, thus, follow-up compounds based on these interactions are expected to be more successful in the drug development process[6]. Consequently, with the increased interest in obtaining RT structures[7], an increasing number of synchrotron beamlines enable RT data collection[8–12]. Indeed, recent structure determinations of ligand-protein structures at RT tend to reveal different conformations of both protein and/or ligands compared to cryo[13–16]. Additionally, fragment screening experiments conducted at RT have revealed significantly fewer fragments binding at RT than at cryo[15,16]. For example, Milano et al.[14] demonstrated differences in ligand binding by comparing a RT structure with a cryogenic structure. Two ligands had identical binding poses at cryo while they exhibited drastically different potencies in cell-based assays. In contrast, at RT the structure of the low potency inhibitor exhibited an extended conformation, indicating a higher flexibility of this ligand that was hidden in the cryo structure. In another example, Huang et al.[13] investigated the binding of a fragment to endothiapepsin as function of temperature. They observed a change of the binding site of a ligand within the active site of the protein at higher temperatures (above 267 K).

In a more systematic study Mehlman et al.[15] partially repeated a fragment screen previously conducted at cryo on the phosphatase PTP1B. For a few fragments they observed subtle conformational changes in ligand binding between RT and cryo. In addition, they observed RT-only binding of one fragment to a different site to that found at cryo, likely caused by small conformational changes preventing binding to this site at cryogenic temperatures. In another study, Dunge et al.[16] compared data from eight different protein/ligand complexes obtained by serial crystallography with data previously collected at cryo. In all cases except one, in which the ligand was only observed at cryo, the binding modes were unchanged. A limitation common to both studies[15,16] is that they did not screen a larger, identical set of compounds at the two temperatures, which thus hampers a thorough comparison of differences in ligand binding.

A general drawback of RT X-ray diffraction experiments is the more than 100-times higher susceptibility of the protein crystals to radiation damage compared to cryogenic data collection, often leading to a significant loss of resolution in the diffraction experiments[17–19]. An attractive alternative method compared to conventional crystallographic rotation data collection from one or a few relatively large protein crystals, is serial X-ray crystallography (SX)[20]. In SX, diffraction still images from hundreds to many thousands of crystals in random orientation are recorded, which are then merged into one complete dataset suitable for structure refinement. By spreading the deposited photon energy over many crystals, the dose and radiation damage effects can be kept at a minimum. Therefore, SX appears well suited for fragment screening experiments at RT as the achievable resolution is key to precise determination of the fragment binding poses.

The main goal of this study is to systematically investigate whether fragment screening at RT can provide additional information on ligand binding relevant for structure-based drug discovery, such as

differences in binding poses, binding sites and conformational changes in the target protein. We further test whether serial synchrotron X-ray crystallography (SSX) is a suitable approach to conduct fragment screening experiments at RT at a resolution that is comparable to conventional cryo-crystallography.

To directly compare RT SSX with conventional cryo data collection, we screen a representative fragment library (F2X entry library), containing 95 molecules, against the enzyme Fosfomycin-resistance protein A from *Klebsiella pneumoniae* (FosAKP)[21]. Fosfomycin, a broad-spectrum antibiotic, is routinely used to treat urinary infections[22]. However, natural resistance can exist in gram-negative bacteria, mainly through the enzyme Fosfomycin-resistance protein A (FosA), that catalyzes the $Mn^{2+}$ and $K^+$-dependent glutathione mediated inactivation of Fosfomycin[23]. Therefore, increased interest exists to identify inhibitors of FosA to enable the use of Fosfomycin to also treat resistant strains[24]. Indeed, inhibitors of FosAKP were recently identified by the screening of a small library of compounds[25], demonstrating the feasibility of this approach.

Here we present the results of our direct comparison of a fragment screen conducted at 296 K and 100 K. To further test the reproducibility of our experiments and to gain significance for our comparisons, the entire screen was repeated twice at 296 K and twice at 100 K. We find that RT serial crystallography leads to data of similar resolution as conventional single crystal data collection at cryogenic temperatures and can be conducted in a highly automated fashion. RT data collection allows us to identify a conformation of the active site of FosAKP, previously not observed in cryo data, offering an additional starting point for drug design. In our screens we identify fewer binders at RT, all binding in the same pose as at cryo. We do not find any fragments that exclusively bind at RT.

## Results
### Serial crystallography enables high-quality data collection at room-temperature

RT-SSX data were recorded using the method of fixed-target crystallography at the HiPhaX instrument installed at beamline P09 at the PETRA III synchrotron in Hamburg, Germany. To enable high-throughput SSX data collection we improved the design of our microporous fixed-target sample holders so that they now provide 12 compartments for different protein/ligand complexes (Fig. 1a)[26]. Crystals were directly grown in the compartments of the sample holders using on-chip crystallization and 3D-printed crystallization chambers (Supplementary Fig. 1a–d). This procedure is based on the commonly applied method of sitting-drop vapor-diffusion and adapted for fixed-target serial crystallography[27]. After crystals grew to sufficient size, the crystallization solution was removed by blotting through the pores of the sample holder and solutions containing the fragments were added to the crystals in the compartment by pipetting (Supplementary Fig. 1e–i). After 24 h incubation time, excess liquid was again removed by blotting through the microporous membranes and a protective cover was slid over each of the sample holders (Supplementary Fig. 1j–m). All sample manipulation steps were carried out in a glove box with high relative humidity ( > 95% r.h.) to prevent crystal dehydration. Data collection was carried out using the Roadrunner sample delivery system permanently installed at HiPhaX (Fig. 1b)[28]. The setup includes a sample chamber which allows the precise control of temperature and r.h. during data collection (7-40 °C, 20-100% r.h.). For this experiment, we selected a temperature of 296 K and 98% r.h. To probe the reproducibility, the SSX screening experiment was performed twice with the same parameters on independently prepared samples (Fig. 2a).

For comparison, the same screen was conducted with conventional single crystal data collection at cryogenic temperatures (100 K) from loop-mounted crystals at beamline P11 at PETRA III. Similar to the RT screen, the cryo screen was also conducted twice using the same

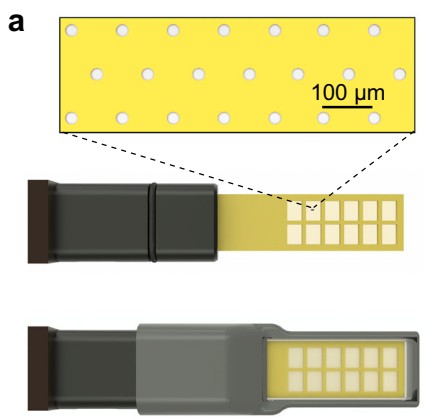

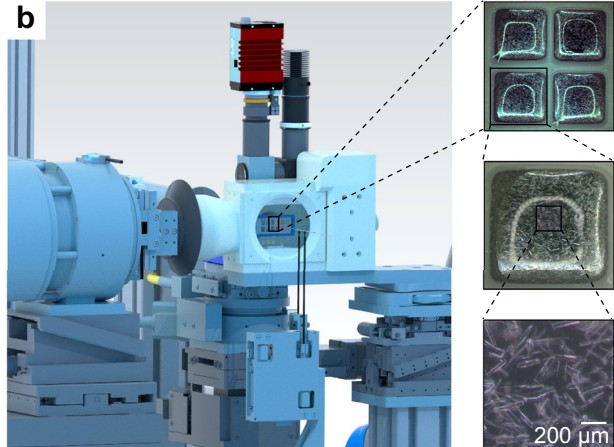

**Fig. 1 | Fixed-target serial crystallography for RT X-ray screening. a** model of the sample holder with microporous Kapton foil (top) attached to a glass fiber-reinforced epoxy frame with 12 compartments for individual compounds. The frame is inserted in a mount with a magnetic base (middle). For data collection the frame is capped with a 3D-printed sleeve with 500 nm thick Mylar windows to protect crystals from dehydration (bottom). **b** view of the sample holder mounted

on the Roadrunner goniometer at the HiPhaX instrument. For data collection the sample holder with its protective sleeve is mounted on the goniometer and then placed into a sample chamber with controlled temperature and relative humidity to prevent crystal dehydration. Insets on the right show FosAKP crystals grown directly on the sample holder with increasing magnification.

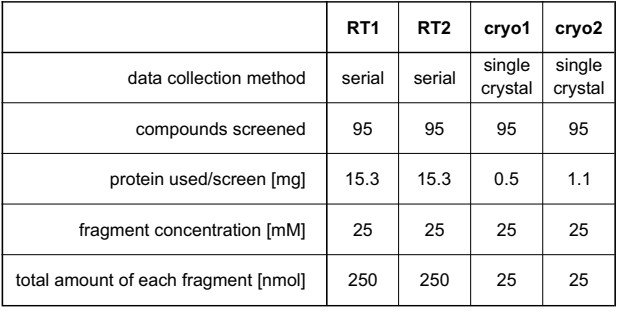

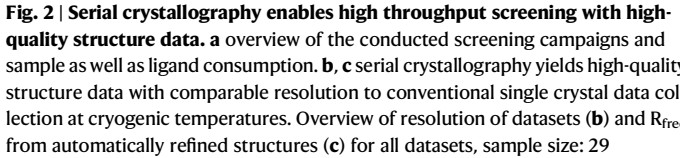

a

|  | RT1 | RT2 | cryo1 | cryo2 |
|---|---|---|---|---|
| data collection method | serial | serial | single crystal | single crystal |
| compounds screened | 95 | 95 | 95 | 95 |
| protein used/screen [mg] | 15.3 | 15.3 | 0.5 | 1.1 |
| fragment concentration [mM] | 25 | 25 | 25 | 25 |
| total amount of each fragment [nmol] | 250 | 250 | 25 | 25 |

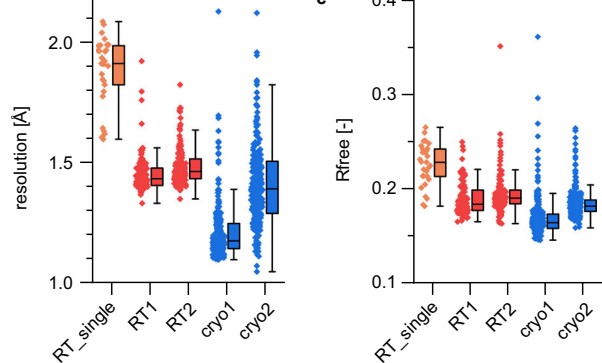

**Fig. 2 | Serial crystallography enables high throughput screening with high-quality structure data. a** overview of the conducted screening campaigns and sample as well as ligand consumption. **b, c** serial crystallography yields high-quality structure data with comparable resolution to conventional single crystal data collection at cryogenic temperatures. Overview of resolution of datasets (**b**) and $R_{free}$ from automatically refined structures (**c**) for all datasets, sample size: 29

(RT_single), 118 (RT1), 143 (RT2), 291 (cryo1), 272 (cryo2). For each screen, individual datapoints (left) are depicted next to summarizing box plots (right) that depict the lower limit of the second quartile range (minimum), the median (center) and the upper limit of the third quartile range (maximum). Whiskers represent 1.5-fold the interquartile range.

parameters with the exception that the mean crystal size was reduced in the second cryo screen. While we used crystals optimized for larger sizes (cryo1), for cryo2 we used the same crystal growth parameters as for the two RT screens, optimized for yielding predominantly smaller crystals. As a result, the average crystal volume for cryo1 was 4.4 times larger than for cryo2 (Supplementary Fig. 2).

To further probe any potential benefit of SSX over conventional single crystal data collection at RT, we collected 29 datasets from eight fragments of the F2X-entry screen (RT_single), including reference apo datasets, by rotation data collection from loop-mounted single crystals, protected by commonly used polymer sleeves to prevent crystal dehydration.

Judging by resolution and refinement statistics both RT datasets are of similar quality as dataset cryo2 (Fig. 2b/c, Supplementary Table 1). The other cryogenic dataset cryo1 exhibited improved quality, which we attribute to the increased crystal size. In contrast to the serial crystallography datasets RT1 and RT2, the control dataset collected from single crystals by rotation data collection at RT (RT_single) is clearly worse, likely due to increased radiation damage.

## Structural difference between RT and cryo

Generally, we observed a slightly larger unit cell volume at RT (Fig. 3a). Interestingly, crystals of the RT screens separated into two distinct unit cell clusters, without any correlation to which screen they belonged. In contrast, both cryo screens exhibited a more homogenous unit cell distribution. Diffraction datasets from both populations showed similar resolution (Supplementary Fig. 3a). Shrinking of the unit cell is accompanied by a small rotation of the FosAKP homodimers against each other (Supplementary Fig. 3b).

A comparison of the ligand-free ("apo") structures from the RT and cryo data shows only small overall differences (0.832 Å RMSD for Cα). However, two distinct differences appear at the two active sites of the homodimer of FosAKP (Fig. 3b). While for the cryo structure a more open active site I is observed, for the RT structure a closed conformation is found. This closure is caused by a shift of the loop consisting of residues 92-101 towards the Mn atom. This loop contains the residues that typically bind potassium, and is therefore also called the K⁺-binding loop[29]. In active site II, this loop is shifted away from the Mn in the RT structure, resulting in an opening of the site and thereby

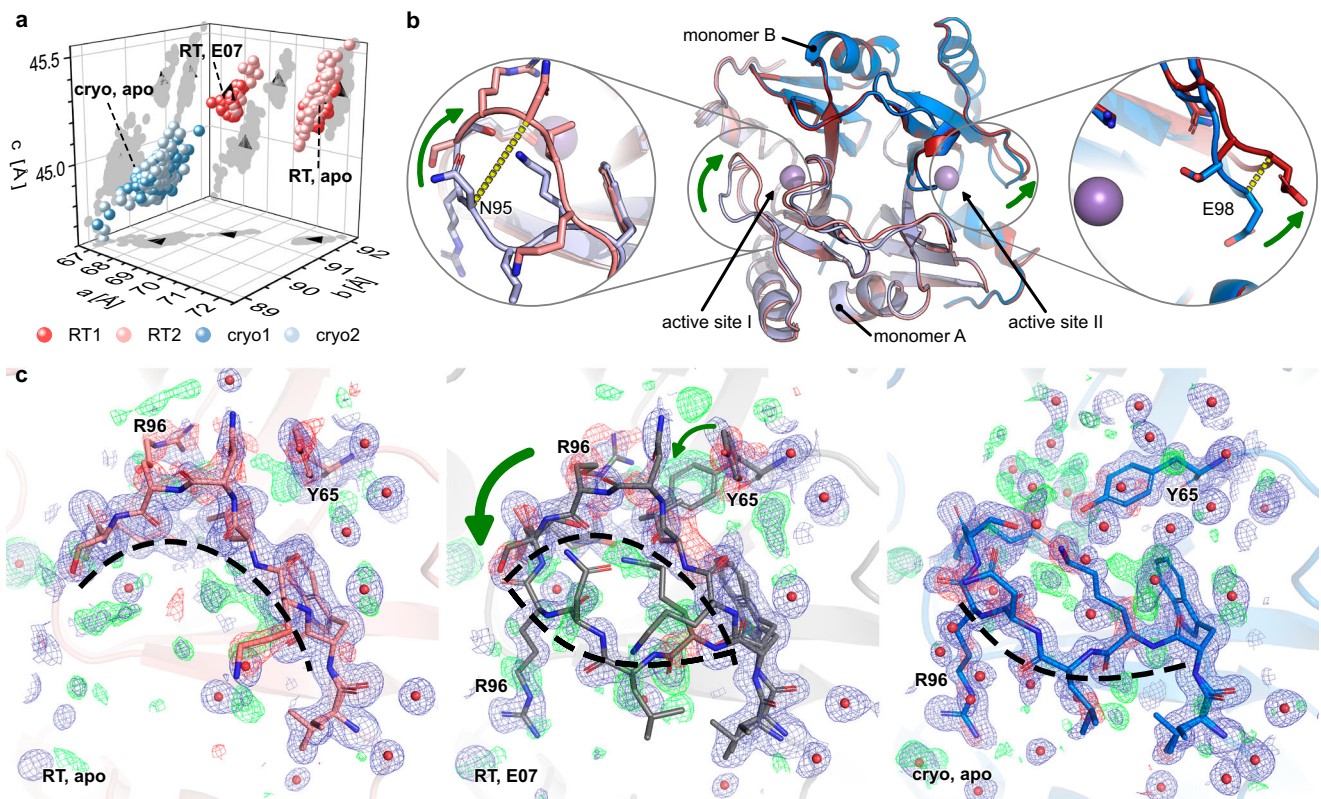

**Fig. 3 | RT crystal structure reveals changes in active site of FosAKP. a** the unit cells of all datasets distribute into three distinct clusters. Representative structures for each cluster are labeled. **b** middle panel, compared to the structure solved at cryo (blue), the RT structure (red) reveals changes in the conformation of the potassium-binding (K+-) loop, directly adjacent to the central manganese ion (purple sphere). Active site I (left panel) closes by up to 7 Å (Cα position of N95). At the same time the second K+-loop moves away from the other manganese, further increasing access to the active site. The Cα of E98 moves by 3 Å. Monomer A is drawn in lighter colors, monomer B in darker colors. **c**, marked differences are observed for the potassium-binding loop and Y65 in one monomer of the FosAKP homodimer. While this loop is in a closed conformation in the RT apo structure (left panel), it is open in the cryo apo (right panel). Interestingly the structure of E07 at RT shows evidence for the presence of both conformations in the electron density (blue mesh for 2mFo-DFc at 1 σ level, green (positive) and red (negative) mesh for mFo-DFc at 3 σ level, both carved at 1.6 Å around model positions of both loop conformations and waters, shown as red spheres, within 4 Å of loop).

in a better substrate accessibility. This loop is already known to undergo conformational changes upon ligand binding[29]. Interestingly, in both apo structures we observe residual difference density in this region (Fig. 3c) and also elevated B factors (Supplementary Fig. 4), indicating high mobility. However, there is only evidence to model a single conformation in the electron density (Fig. 3c). In contrast, some structures from the smaller RT unit cell cluster (Fig. 3a) exhibit sufficiently strong electron-density to model both conformations (Fig. 3c).

### Fragment hits
Over all screens, a total of 31 out of 95 fragments are observed binding to FosAKP. The fragments bind at 7 different sites to the protein (Fig. 4a/b and Supplementary Fig. 5/6). Because sites 5-7 are located between protein molecules at crystal contacts, which are likely not affecting substrate entry or binding to the active site, we consider them pharmaceutically less relevant. In contrast, sites 1-4 are either in the active site or close by and have a higher potential to influence the enzyme's activity (Fig. 4c). In total, 7 fragments bind in the two RT screens, of which 3 bind to pharmaceutically relevant sites 1, 2 and 4 (Fig. 4a, d–f). In contrast, 31 fragments are observed binding in the cryo screens, but only 6 of these bind to the more interesting sites 1-4 (Fig. 4b, d–f).

### Fragments binding at RT and cryo temperature
As seen from Fig. 4e, there are two fragments that bind near an active site that were observed in both RT screens and both cryo screens and one further fragment that showed up in all screens except RT2. One of these, fragment A09, binds covalently to Cys126 near active site I and is further stabilized by a hydrogen bond at fragment binding site 1 (Figs. 4a and 5a–d). The binding poses at both temperatures are highly similar (Fig. 5c). Covalent binding of isothiazolones like A09 has previously been described[30]. A more detailed discussion of the chemical interactions of the fragments with the protein is provided in Supplementary Notes 1. The other fragment seen in all screens, A12, binds identically at both temperatures non-covalently to the active site II via three hydrogen bonds plus coordination of the central manganese ion at site 4 (Figs. 4a and 5e–h). Finally, fragment G08 binds similarly at both temperatures non-covalently further away from the active site at binding site 2 mainly via two hydrogen bonds to the backbone of the protein (Figs. 4a and 5i–l). This fragment was not seen to bind in the RT2 screen, probably caused by experimental uncertainties (Fig. 4f).

### Fragments exclusively identified at cryo
In addition to the previous three fragments, we identified three more fragments (A06, B02 and G12) at binding sites 1-4 that were only observed at cryo temperatures.

Fragment A06 binds in the active site 2 via three hydrogen bonds similar to fragment A12 (site 4, Figs. 4b and 6a–d). Fragment B02 binds at the same site as A06 and A12 via manganese coordination and a hydrogen bond (site 4, Figs. 4b and 6e–h). Finally, fragment G12 binds at two sites (site 3 and 7) of which only binding site 3 is considered as relevant for further drug development and is therefore discussed in

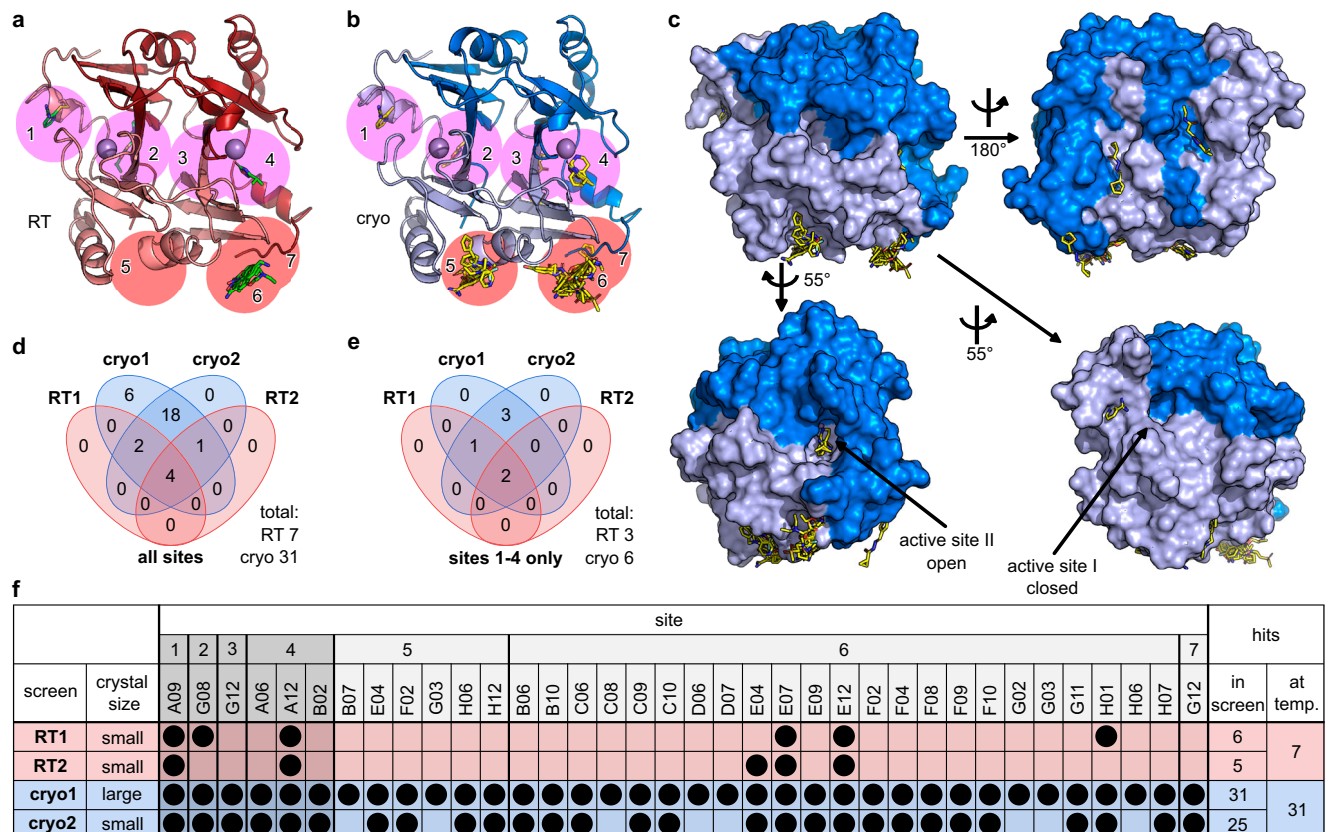

**Fig. 4 | X-ray fragment screen at RT vs cryo reveals different amounts of bound ligands. a** ligands (green sticks) identified at RT by serial crystallography with soaked ligands. The two FosAKP protomer subunits are colored in light and dark color (see also Fig. 3 legend). **b** ligands (yellow sticks) identified at cryo by single crystal diffraction with soaked ligands. Binding sites are numbered and sites at or near the active site of FosAKP are highlighted with purple circles while sites at surface pockets and mediated through crystal contacts are highlighted with red circles. **c** fragments bind across the surface of FosAKP (top panels). A few fragments are identified either inside the open active site I (lower right panel) or close to the closed active site II (lower left panel). **d, e** overlaps of identified hits between the four different screens for all hits (**d**) and for sites 1–4 only (**e**). **f** overview of individual fragments identified in the four screens as hits sorted by binding site. Fragments binding at more than one site (E04, F02, G03, G12, H06) are listed for each site. The pharmceutically interesting sites 1-4 are shaded darker.

detail (Figs. 4b and 6i–l). At site 3 the fragment forms two hydrogen bonds with backbone atoms of FosAKP.

### Further fragments observed at RT and cryo

The majority of bound fragments is observed at binding site 6 (Fig. 4a/b and Supplementary Fig. 7a-p) which is located at the interface between two dimers generated by crystallographic symmetry (Supplementary Fig. 7q and r). In each dataset RT1 and RT2 three fragments are found at this site. In strong contrast, in the screens cryo1 and cryo2, 23 and 17 fragments are identified at this site. An analysis of the width of this cleft between the two dimers only shows a small reduction at cryo compared to RT, which might contribute to the increased visibility of ligands at this site (Supplementary Fig. 7r and s). An interaction common to all except one fragment is the cation-π stacking of the guanidyl group of Arg33 of chain A with an aromatic ring of the fragments (Supplementary Fig. 7d, h, i, p). For the fragments observed at both temperatures, again, there are no obvious differences in the binding poses (Supplementary Fig. 7c, g, k, o).

## Discussion

Here we demonstrated that our method of fixed-target serial crystallography in combination with on-chip crystallization and multi-sample holders is well suited for fragment screening experiments at RT. The resolution we achieved with this method at RT is almost the same as for conventional single crystal data collection at cryo, highlighting the potential of serial crystallography for measurements at near

physiological temperatures. In contrast to serial data collection, the achievable resolution of conventional data collection at RT using single crystals is reduced by at least 0.4 Å.

Our RT structure of FosAKP reveals a previously undiscovered conformation of the K+-binding loop in the active site of the protein, likely modulating enzymatic activity by restricting access to the central Mn2+ ion. Notably, this state is not observed in any of the available cryo structures including ours and is also not predicted by AlphaFold3 (Supplementary Fig. 8 and Supplementary Notes), presumably hinting at a current strong prediction bias of the algorithm towards cryogenic temperatures. This is likely caused by the training data from the PDB of which as of July 2025 about 94% are derived from structures collected at temperatures below 220 K. Moreover, cryo structures of FosAKP were likely included in the training dataset. The different conformation of FosAKP observed at RT could provide a valuable alternative starting point for new molecular designs for a difficult-to-target system as underscored by the generally low number of hits at cryo and RT at or near the active site.

In our comprehensive screening experiments with 95 different fragments, at RT we observed 3 out of the 6 ligands that were identified at cryo and which bound to sites that are presumably of pharmaceutical relevance, and in total 7 out of 31 ligands that bound to these sites. The differences in identified ligands between the reproduced screenings at each temperature highlights the general challenge in fragment screening associated with the low signals from the weakly binding fragments. While at cryo the stronger data from the larger crystals of

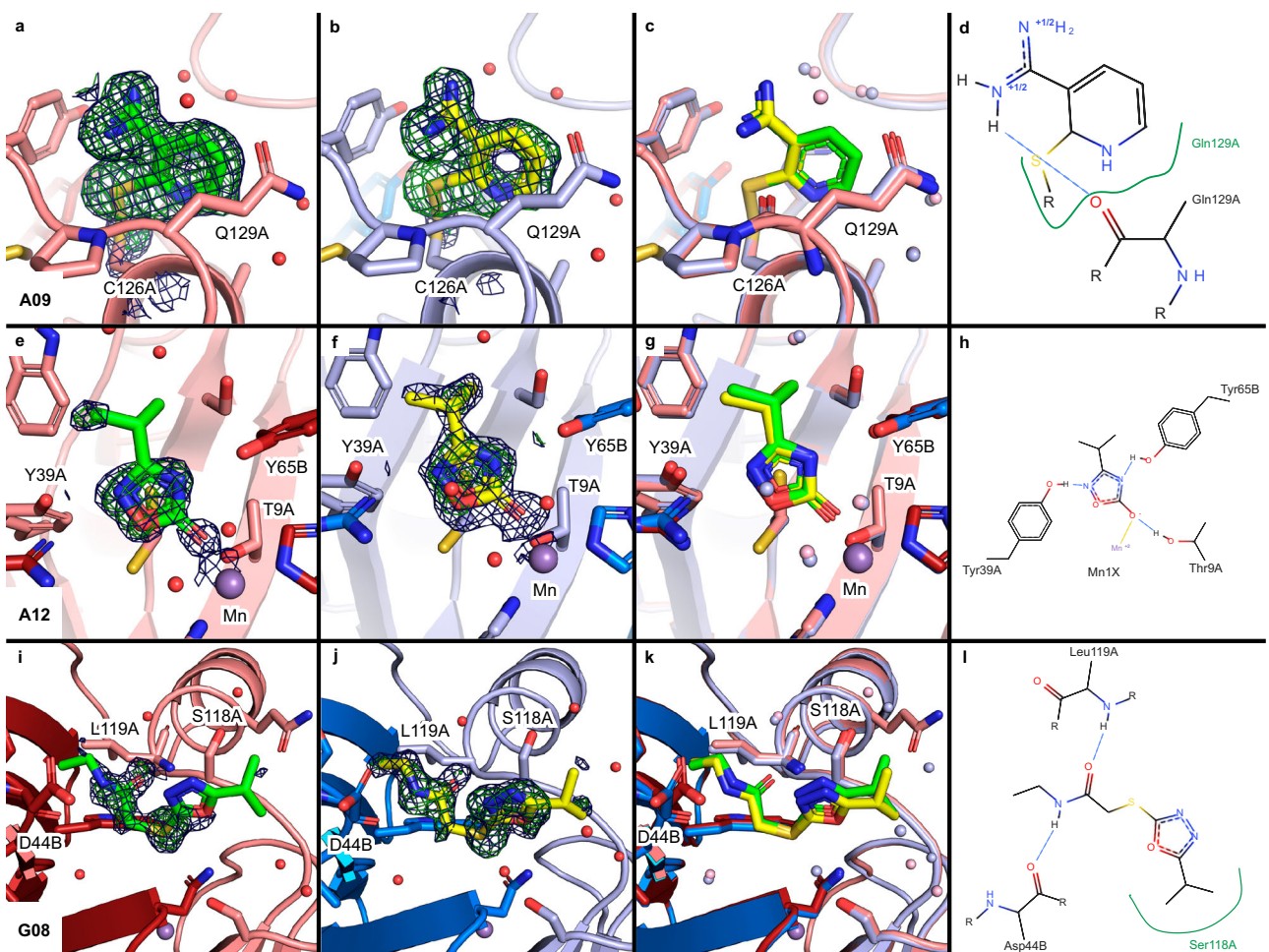

**Fig. 5 | Fragments identified through RT and cryogenic data collection binding at or near the active site of FosAKP.** Refined RT structures (**a**, **e**, **i**) and cryo structures (**b**, **f**, **j**) are shown with PanDDA event map (blue mesh, 2 σ level) and Z-map (green mesh, 3 σ level) drawn around the ligand (carved at 2 Å), coloring as in previous figures. Protein residues within 4 Å of ligands are depicted as sticks, water molecules as red spheres. An overlay of structures from both temperatures is shown in (**c**, **g**, **k**) (water molecules of RT structures in light red and of cryo structures in light blue). Interaction diagrams of ligands with the protein are shown in (**d**, **h**, **l**) generated using PoseEdit[45]. Protein residues interacting with the ligands are labeled. Hydrogen bonds are shown as dashed, blue lines. Green lines indicate hydrophobic interactions, yellow dashed line metal coordination.

cryo1 compared to cryo2 can explain the higher hit rate, the differences between the RT screens derived from identical crystal sizes point to still uncontrolled parameters hindering a better reproducibility. The hit identification method PanDDA is depending on the homogeneity of the analyzed data, and preclustering of datasets, as was also done here, improves this[15,31]. Some of the identified ligands were likely borderline cases, the detection of which depends on the generated reference map in PanDDA. Therefore, subtle differences in diffraction quality and homogeneity of data between RT1 and RT2 could lead to a failure, for example, in identifying ligand H01 in both screens. This emphasizes the need for more systematic studies such as ours allowing to derive more general conclusions.

Since all sample preparation steps including protein crystallization and fragment application are conducted at RT, we expect that ligand binding generally also occurs at RT and is not induced by the cooling process. This implies that all ligands observed at cryo must already be present at this binding site at RT. Their apparent absence in the RT data can most likely be explained by increased thermal motion and disorder resulting in the loss of visibility in the electron-density. This explanation is supported by previous observations by Milano et al. and Huang et al.[13,14] who also observed an increased flexibility resulting in different or multiple ligand conformations at RT compared to cryo.

Vice versa all ligands observed at RT are also visible in identical binding poses at cryo, clearly indicating that flash-cooling does not induce any artifacts nor any loss of ligand binding. From this we conclude that ligands only observed at cryo exhibit increased disorder at RT and any specific interactions exclusively observed at cryo are weaker at RT. In contrast, we expect that ligands observed in well-defined binding poses at RT are of much higher relevance since they exhibit proven interactions at near physiological temperatures. Therefore, these ligands should form a more suitable basis for downstream drug development and should be prioritized.

The requirement for larger amounts of both protein and ligands for RT screening experiments might represent a limitation for certain screening projects (Fig. 2a). In particular for these cases, we propose a workflow of first conducting a conventional single crystal X-ray screen at cryo followed by an RT screen of the previously identified ligands using our method of fixed-target serial crystallography.

In other cases where the availability of neither protein nor ligands is limited, our method has the big advantage of simplified sample preparation and handling. By combining RT data collection with on-chip crystallization, on-chip ligand application, and multi-compartment fixed-target sample holders, our method avoids the most time-consuming steps required for conventional screening

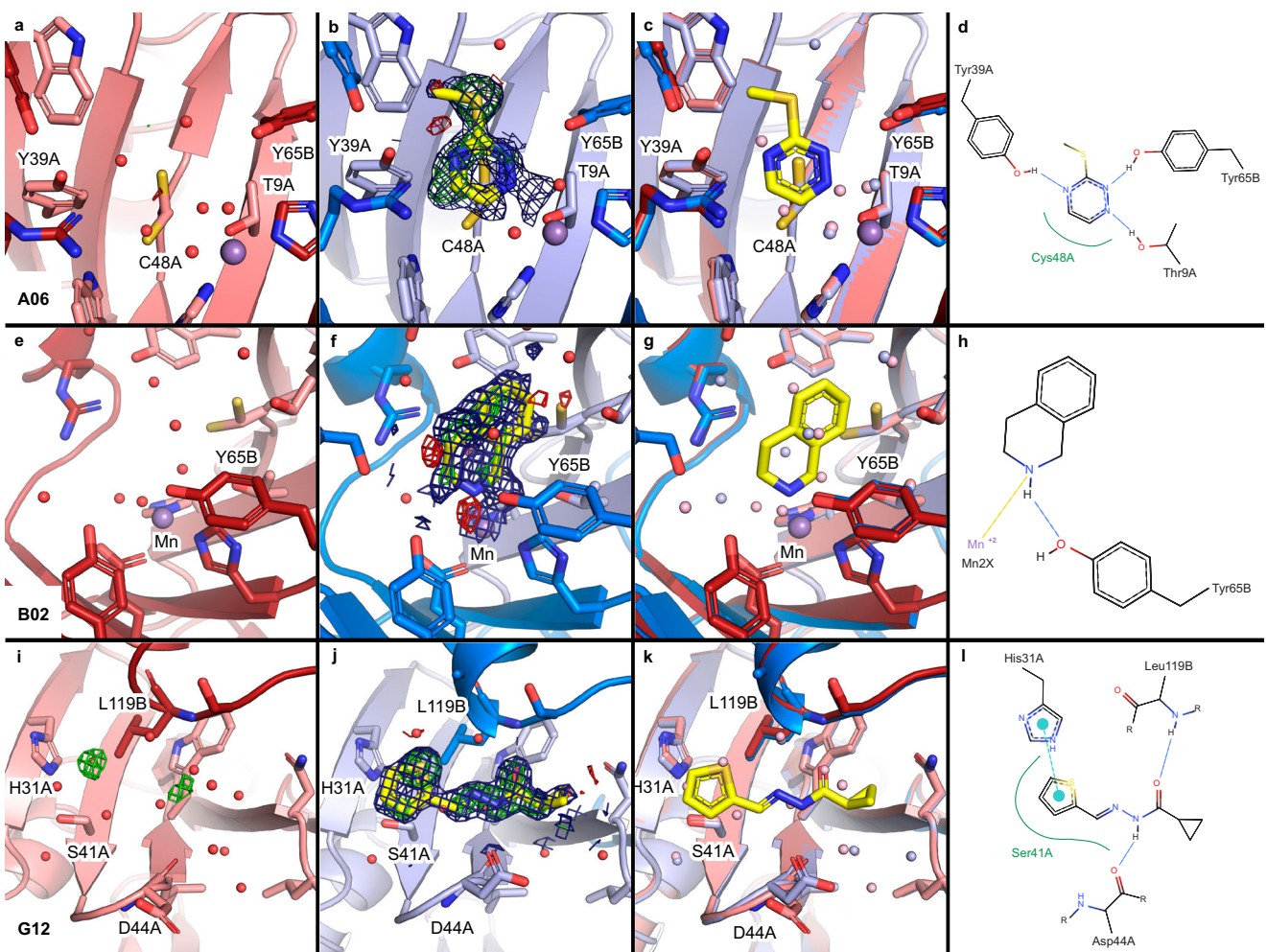

**Fig. 6 | Fragments identified only through cryo data collection binding at or near the active site of FosAKP.** For comparison, automatically refined RT structures from samples with the same compound are shown with Z-map around the expected position taken from the cryo structure (**a**, **e**, **i**). Cryo structures (**b**, **f**, **j**), are shown with PanDDA event map (blue mesh, 2 σ level) and Z-map (green mesh, 3 σ level) drawn around the ligand (carved at 2 Å), coloring as in previous figures. Comparison of ligand binding sites with all residues within 4 Å of ligand are shown

as sticks. Water molecules are colored lightred (RT) and lightblue (cryo) (**c**, **g**, **k**). Schematic interactions of ligands with FosAKP are shown in the (**d**, **h**, **l**) generated using PoseEdit[45]. Protein residues interacting with the ligands are labeled. Hydrogen bonds are shown as dashed, blue lines. Green lines indicate hydrophobic interactions, yellow dashed line metal coordination, and cyan dashed line pi-pi interactions.

experiments. These include the search for suitable cryo-protectant, soaking individual crystals with ligands, loop-mounting individual crystals, flash-cooling, as well as mounting of the individual crystals on the goniometer and centering in the X-ray beam. While crystal growth on the sample holders, ligand application, blotting, cover sleeve application and mounting on the beamline were conducted manually in the present study, it has to be noted that each sample holder carries 12 different samples and blotting, sleeve application and mounting only need to be done once per sample holder, thereby increasing throughput. Moreover, we are working on automation of these steps to further reduce manual intervention and increase reproducibility as well.

In situ data collection directly in crystallization plates is another approach for RT screening that circumvents the need for mounting individual crystals[8,15]. Typically, multiple crystals are merged together to obtain complete datasets. This can also help to reduce the detrimental effect of radiation damage that limits resolution from single crystals at RT and improve the overall resolution. A potential drawback compared to the serial data collection presented in this study is the increased scattering background originating from the crystallization plate and the crystallization droplet.

In particular at X-ray free electron lasers, high-viscosity extruders are used for sample delivery in serial crystallography experiments. Based on this method multi-sample devices have been developed[32], however at the moment these do not provide a high enough throughput for larger screening campaigns. Furthermore, the background from the sample medium could be a concern reducing data quality.

In our RT experiments using monochromatic radiation the measurement time per compound was about 30 min at a data collection rate of 25 Hz. Conducting these experiments at synchrotron beamlines with much higher X-ray intensities or even polychromatic X-rays[33,34] should enable data collection with frame rates of up to 1 kHz or faster[35]. This will reduce the time required for measuring one compound to less than one minute, allowing the screening of up to 1200 compounds per 24 h. Such a high throughput in combination with significantly reduced sample preparation efforts will enable the use of serial crystallography at RT as a primary X-ray screening method. The resulting structural information of protein-ligand interactions with near-atomic resolution at near-physiological conditions will directly facilitate ligand optimization and thereby streamline structure-based drug development.

## Methods

### Protein expression

Fosfomycin-resistance protein A from *Klebsiella pneumoniae* (FosAKP) was expressed and purified as described[29]. Briefly, FosAKP containing a C-terminal His6-tag was expressed in *E. coli* strain BL21(DE3)pLysS. A shaker flask with TB medium was inoculated 1/100 (v/v) with an overnight culture. Cells were grown at 37 °C until an OD600nm of 0.6 was reached. Expression was induced with 0.1 mM IPTG, and culture was further incubated at 18 °C overnight. Cells were harvested by centrifugation (5000 g, 30 min) and lysed in 0.1 M NaPhosphate, pH 7.5 by sonication. After centrifugation (50000 g, 45 min) supernatant was passed over Ni-NTA agarose, eluted with imidazole, concentrated and further purified by size-exclusion chromatography in 10 mM Hepes, pH 7.5, 75 mM NaCl. Protein was concentrated to 42 mg/mL, flash-frozen in liquid nitrogen and stored at -80 °C until further use.

### F2X entry library

Compounds H09 and H10 of the original F2X entry screen[21] were not available at the time the screen was established on our site. Instead we added another fragment, 4-brom-1H-pyrazol. For this study this led to a combined number of 95 fragments in the F2X entry screen compared to the 96 fragments of the original study. The fragments were acquired from various vendors (for details incl. order and lot numbers see Supplementary Data 1). No further characterization was conducted.

### SX sample preparation

For efficient SX fragment screening at room-temperature data collection, we have developed a fixed-target sample holder equipped with a micro-perforated Polyimide membrane, subdivided into 12 physically separated compartments using a plastic frame (Fig. 1a). Each compartment has a size of $4.5 \times 4.5$ mm$^2$, sufficient to accommodate several ten thousand crystals per compartment (Fig. 1b). For the experiments FosAKP was directly crystallized on these sample holders[27]. 25 mg/mL FosAKP in 10 mM Hepes, pH 7.5, 75 mM NaCl was first supplemented with a final concentration of 6 mM MnCl$_2$ and then mixed with an equal volume of 16% (w/v) PEG3350, 0.25 M MgCl$_2$, 0.2 M KBr, 0.1 M BisTris, pH 5.5 and 1/10 volume of seed stock in 26% (w/v) PEG3350, 0.25 M MgCl$_2$, 0.2 M KBr, 0.1 M BisTris, pH 5.5. Of this solution approximately 14 μL were added per window of the fixed-target chip. The sample holder was then inserted into a 3D-printed crystal growth chamber with 3 mL of precipitant solution in the bottom for vapor-diffusion crystallization and incubated at 20 °C.

For fragment application sample holders were removed from the crystal growth chamber and excess precipitant was removed by blotting through the micropores of the membranes, before 10 μL of fragment solution at a concentration of 25 mM in 5% DMSO were pipetted to the crystals in the individual compartments. Sample holders were then placed back into the growths vessel and incubated for 24 h. Before data collection blotting was repeated for removal of excess liquid in order to minimize background scattering. Sample holders were then equipped with a protective cover to prevent them from drying-out and stored in a humid atmosphere (Fig. 1a). Compound addition and liquid removal were conducted in a glove box with >95% rel. humidity.

### SX RT data collection

Room-temperature fragment screening experiments using SX were performed at DESY's recently established experiment for High-throughput Pharmaceutical X-ray screening (HiPhaX) at the PETRA III storage ring in Hamburg. The beamline is equipped with a Roadrunner goniometer[28] for fast raster scanning of samples which are immobilized on a fixed target. It further provides a measurement chamber with precisely controlled temperature and relative humidity for well-defined data collection conditions.

SX diffraction data were collected at an X-ray energy of 16 keV/ 0.7749 Å on a Pilatus 6 M detector. The X-ray spot size at the sample position was approximately $20 \times 20$ μm$^2$ FWHM with a photon flux of $3.5 \times 10^{11}$ ph/s. Sample holders were scanned in horizontal direction through the X-ray beam in fly scan mode at a speed of 0.375 mm/s and a frame rate of 25 Hz corresponding to an exposed area of about $35 \times 20$ μm$^2$ (v×h). With a typical crystal size of $20 \times 60 \times 300$ μm$^3$ this corresponds to a dose of about 4 kGy per exposure. Using a line spacing of 40 μm about 34,000 diffraction images were collected per compartment. Each line was collected at a fixed angle of the sample holder with respect to the X-ray beam. After scanning of each line this angle was changed incrementally, resulting in a total rotation range of 30 degrees for each compartment, thereby reducing the risk of incomplete data due to preferred orientation of the crystals on the sample holder. The 12 compartments of each sample holder were measured sequentially. Data collection time was about 30 min per compartment.

### SX data processing

The data were processed using CrystFEL (v0.10.1)[36]. The peakfinder8 algorithm was used for identifying the Bragg peaks with parameters: --min-snr=8 --threshold=5 --min-pix-count=1. Initially, images were indexed using mosflm without prior information about unit cell parameters[37]. Datasets exhibiting clear presence of two unit-cells were separated according to the *a* axis. Subsequently, xgandalf[38] was used for indexing using the previously determined unit cell for each dataset, --multi option and integrated with −int-radius=3,6,8. Scaling and merging of the data into point group mmm was carried out applying partialator in CrystFEL, using three iterations and --push-res=1.0 and figures of merit were calculated using compare_hkl and check_hkl, all part of the CrystFEL package. MTZ files for crystallographic data processing were generated from CrystFEL merged reflection data files using F2MTZ of the CCP4 program suite[39]. The high-resolution limit was determined as the resolution where I/σI falls below 1. For this the merging statistics were calculated for a range of resolution cut-offs in 0.1 Å steps and the one closest to 1 was selected. Structure refinements were conducted using an established pipeline based on phenix.refine[4,40]. At this point no fragments were modeled yet. Resulting structures and map files were analyzed and clustered using cluster4x[31], which was in particular useful for the RT datasets with two unit cell clusters. PanDDA was run on each cluster for hit identification[41]. Hits were subsequently refined using phenix.refine and manual model building in Coot and deposited in the PDB (Supplementary Table 2)[40,42]. The initial occupancy of the fragments was set to 2*(1-BDC) and subsequently refined during model refinement (Supplementary Table 2)[41]. For the RT apo structure 9 windows from one chip were combined to increase the achievable resolution. To enable a better comparison of data processing statistics with the single crystal datasets, a more precise resolution limit was estimated by linear interpolation to determine the resolution at which I/σI = 1.

A common method to detect weak signals in serial crystallographic datasets, in particular in time-resolved studies, is the use of isomorphous difference maps (Fo-Fo maps). For the fragments identified by PanDDA we checked the signal for the fragments in the data by calculating q-weighted Fo-Fo maps[43]. For datasets with an *a* axis > 70.5 Å we used the FAKP apo dataset as reference. Because there was no FosAKP apo dataset in the cluster with the smaller unit cell (*a* axis ≤ 70.5 Å), we selected a dataset from this cluster that presumably did not have a bound fragment. For this, we sorted all datasets from this cluster based on their Rfree value and filtered out any datasets from fragments that had been identified in the cryo screens. This led to D03 (dataset xg-new_FAKP_F2X_chipD_grid_fly_001_window_9) as reference dataset for the second cluster. We used Xtrapol8 to calculate q-weighted

Fo-Fo maps[43]. Additionally we calculated extrapolated structure factors from these and derived 2Fextrapolated-Fc maps that show enhanced signal also for weakly bound ligands[43]. All figures of models were prepare using PyMOL Molecular Graphics System, Version 3.1.

## Single crystal sample preparation

For comparison, conventional X-ray rotation data collection at 100 K from single crystals was performed. Here, we followed two approaches for sample preparation. For the soaking experiments, the protein was first crystallized by mixing 0.45 μL of 12 mg/mL (cryo1) or 25 mg/mL (cryo2) protein solution in 10 mM Hepes, pH 7.5, 50 mM NaCl, with 0.45 μL 16% (w/v) PEG3350, 0.25 M MgCl$_2$, 0.2 M KBr, 0.1 M BisTris, pH 5.5 and 0.1 μL crystal microseeds in 26% (w/v) PEG3350, 0.25 M MgCl$_2$, 0.2 M KBr, 0.1 M BisTris, pH 5.5. After at least 4 days of crystal growth, 40 nL of 500 mM compounds, solubilized in 100% DMSO, were added using an acoustic droplet dispensing system. After fragment soaking for at least 24 h, crystals were briefly soaked in 16% (w/v) PEG3350, 0.25 M MgCl$_2$, 0.2 M KBr, 0.1 M BisTris, pH 5.5, 5% (v/v) DMSO, 10.5% EG without fragment for cryo protection, and afterwards directly flash cooled and stored in liquid nitrogen until data collection.

## Single crystal cryogenic data collection

Cryogenic X-ray diffraction data were collected at a temperature of 100 K from loop-mounted single crystals at beamline P11 at PETRA III using an X-ray energy of 12 keV/1.0332 Å (cryo1) or 15 keV/0.8266 Å (cryo2) on an Eiger 16 M detector. The beam size was adjusted to $50 \times 50$ μm$^2$, with an attenuated photon flux of $2 \times 10^{11}$ ph/s. The average crystal size was about $20 \times 60 \times 300$ μm$^3$. With 1800 images collected over 360 degrees total rotation with 10 ms exposure per image, this results in an average dose of 14 MGy per crystal.

Data was processed using XDS within autoproc. Similar to the RT datasets, the resolution limit was set to the point where I/σI falls below 1. Structure refinements and subsequent hit identification were performed using the same procedure as for the serial data.

Hits overlapping with the RT datasets and those at sites 1–4 were further refined using phenix.refine and manual model building in Coot and deposited in the PDB (Supplementary Table 2)[40,42]. For the cryo apo-structure the data from 3 crystals were combined to increase the achievable resolution.

## Single crystal RT data collection

For a direct comparison between our method of SX data collection with rotation data collection from loop-mounted single crystals at cryo and our method of serial data collection at RT, we performed reference measurements using conventional data collection at RT at HiPhaX. For this, crystals were prepared identical to the cryo data collection. The crystals were subsequently mounted on Kapton loops and protected from dehydration by using a polyethylene terephthalate sleeve, filled at one end with the crystallization solution (Mitegen, USA). Like the SX RT data, these data were collected at HiPhaX at 16 keV/0.7749 Å. With the beam attenuated to 10% (approximately $3.5 \times 10^{10}$ ph/s), 1200 images over 240 degrees with 40 ms exposure per image were collected, amounting to approximately 0.48 MGy per crystal. Data were processed identically to the cryo data.

## Alphafold prediction of FosAKP structures

For comparison of our experimentally determined structures we additionally used AlphaFold3 to predict the structure of FosAKP[3]. Two copies of the sequence of the expressed FosAKP polypeptide together with Mn$^{2+}$ and K$^+$ as cofactors was given as input to the AlphaFold Server. The predicted structures were aligned using GESAMT[44] and RMSD was calculated from superposed structures using all Cα atoms.

## Reporting summary

Further information on research design is available in the Nature Portfolio Reporting Summary linked to this article.

## Data availability

All structure models and structure factors, including PanDDA event maps, have been deposited in the PDB with the IDs 9G1A, 9G1B, 9G1C, 9G1D, 9G1E, 9G1F, 9G1G, 9G1H, 9G1I, 9G1J, 9G1K, 9G1L, 9G1M, 9RPX, 9RPY, 9RPZ, 9RQ0, 9RQ1, 9RQ2, 9RQ3, 9RQ4, 9RQ5, 9G1N, 9G1O, 9RQ6, 9G1P, 9RQ7, 9RQ8, 9RQ9, 9RQA, 9RQB, 9RQC, 9RQD, 9G1Q, 9RQE, 9G1R, 9G1S, 9RQF, 9RQG, 9RQH. PDB 5V91 was initially used as starting model for structure determination. All models and structure factor files derived from the automatic refinement pipeline and described in this manuscript are deposited to Zenodo (https://doi.org/10.5281/zenodo.15863149). The results from the PanDDA analysis used as starting points for the deposited structures and isomorphous difference and extrapolated structure factor maps for the RT hit fragments are provided there as well. Source data for Figs. 2b/c, 3a and Supplementary Figs. 2c, 3a, 7s are provided with the paper as a Source Data file. Source data are provided with this paper.

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

## Acknowledgements

We acknowledge DESY (Hamburg, Germany), a member of the Helmholtz Association HGF, for the provision of experimental facilities. Room-temperature data collection was carried out at PETRA III beamline P09. Cryo data collection was collected at beamline P11 and we would like to thank Johanna Hakanpaeae, Helena Taberman, Guillaume Pompidor and Spyros Chatziefthymiou for assistance. Beamtime was allocated for BAG proposal BAG-20230011. This research was supported in part through the Maxwell computational resources operated at Deutsches Elektronen-Synchrotron DESY, Hamburg, Germany. The authors further acknowledge financial support from the Federal Ministry of Education and Research (BMBF) via the Röntgen-Ångström-Cluster project "X-ray drug design platform" (13K22CHB, AM) and project "conSCIENCE" (16GW0277, AM). This work was also supported by the Helmholtz Society through the projects FISCOV (AM), FISVIR (AM) and SFragX (AM) and the Helmholtz Association Impulse and Networking funds InternLabs-0011-HIR3X (AM/HNC). Further we acknowledge funding by the Cluster of Excellence "CUI: Advanced Imaging of Matter" of the Deutsche Forschungsgemeinschaft (DFG)—EXC 2056—project ID 390715994 (HNC). We also thank Oleksandr Yefanov for helping with establishing a data processing pipeline and Tobias Krojer for providing scripts to facilitate PDB deposition of PanDDA event maps. We thank Tim Pakendorf for help with CAD images and T.J. Lane and Winfried Hinrichs for critical comments on the manuscript.

## Author contributions

Conceptualization: S.G., A.M. Sample preparation, data analysis, structure refinement: S.G. Serial data processing: S.G., M.G. Data collection: S.G., P.F., S.F., P.Y.A.R., S.T.V., A.C.R., J.S., J.M. Technical development: S.G., P.F., L.G., J.M., A.M. Preparation of microporous sample holder: D.E., M.B. Funding acquisition: H.N.C., A.M. Manuscript writing–original draft: S.G., A.M. Manuscript writing–review and editing: S.G., S.F., H.N.C., M.B., A.M.

## Funding

## Competing interests

PF and AM are shareholders of suna precision GmbH, which developed and manufactured parts of the hardware. The other authors declare no competing interests.
