## [Transparent Peer Review file · Nature Communications]

Room-temperature X-ray fragment screening with serial crystallography

Corresponding Author: Dr Sebastian Günther

Version 0:

Reviewer comments:

Reviewer #1

(Remarks to the Author)

The authors present a method for room temperature fragment screening using a serial X-ray crystallography approach. The unique aspect of this method is the use of a microporous solid-target sample chip whose design has been improved for this specific application and a specialized data collection chamber that keeps constant temperature and humidity. A library of 95 small molecule fragments is screened against the FosA protein from *Klebsiella pneumoniae* at room temperature and with standard cryogenic approaches, in replicate to enable systematic comparison. This led to the identification of 31 fragment hits binding in 8 distinct binding sites. Only 7 out of the 31 hits were identified at room temperature and for those common at both temperatures binding sites and binding poses were identical. All the hits identified at RT were also found under cryogenic conditions. However, an alternate conformation of the potassium binding loop was also found at RT. The authors attribute differences in ligand binding to increased thermal motion and disorder at room temperature, highlighting its potential relevance for drug design. The manuscript is well written and concise establishes proof of concept for the method and presents valuable results.

However, in this reviewer's view, a number of points need to be addressed:

1. The abstract, lines 33-35, states: "But overall. We observed more binders at cryo, however mainly at physiologically likely less relevant sites"

I find this phrase could be misinterpreted. It might be understood that cryogenic experiments might not have identified pharmacologically relevant or all pharmacological relevant sites in this experiment. But the results show that all fragment hits identified at RT were also identified at cryogenic temperature. Moreover, cryogenic experiments identified more fragment hits at pharmacologically relevant sites than the RT approach. Please, revise this phrase to clarify. For example: "We observed more binders at cryo, both at physiologically relevant and non-relevant sites"

2. Figure 2 Panel B. RT1 and RT2 data has been grouped in bins of 0.1 angstroms. Instead of continuous values which gives a strange appearance to the graph. However, this makes comparison with the cryogenic data difficult and no explanation for this is given. Please plot continuous values for RT data as well.

3. In the results section (line 130-145) a description of how the chips are prepared for data collection is introduced, but not of the previous steps. i.e. how crystals are grown and ligands delivered to the crystals, please include a succinct description of the whole protocol including these steps in the results section. This will facilitate understanding both by specialist and non-specialist readers.

4. Line 290. "our method avoids the most time-consuming steps required for conventional screening experiments. These include the search for suitable cryo-protectant, soaking individual crystals with ligands, loop-mounting individual crystals, flash-cooling, as well as mounting of the individual crystals on the goniometer and centering in the X-ray beam."

This reviewer disagrees with this statement. In this method, soaking of crystals is still necessary (not single but multiple microcrystals). Moreover, protecting sieves have to be applied to the chips to avoid crystal dehydration once out of the

crystallization chamber, all of which are probably manual operations. Similarly mounting of individual crystals in a goniometer is replaced by mounting of the chips into the temperature-controlled chamber and the necessary preparations for X-ray measurements. In cryogenic screening most of these operations are carried out with robotic equipment and automated sample exchangers that present hundreds of samples to the X-ray beam in fully automated mode making the process very efficient and probably more efficient than the process presented here. Adding to this, the authors do not compare the efficiency of their method with other RT approaches, like for example in situ data collection where samples are measured directly in crystallization plates avoiding crystal manipulations (refs. 8 and 15). While the RT screening method presented here has a value of its own, I disagree with the notion that it is less time consuming or more efficient than existing cryogenic approaches, which are highly automated, at least at this point. Indeed, while measuring collections of 500-1000 fragments is standard with cryogenic data collection, only 95 fragments were measured in this work, which are relatively low numbers in the context of fragment screening. A more balanced comparison of this approach, cryogenic methods and other RT methods previously applied to ligand screening should be presented in the discussion.

5. Only 19 PDB depositions are mentioned linked to this work: 9G1A, 9G1B, 9G1C, 9G1D, 9G1E, 9G1F, 9G1G, 9G1H, 9G1I, 9G1J, 9G1K, 9G1L, 9G1M, 9G1N, 9G1O, 9G1P, 9G1Q, 9G1R, 9G1S. However, the authors report 31 fragment hits were found under cryogenic conditions for which 7 were also found at room temperature. Why not all models for fragment hits have been deposited? One would expect 38 depositions associated to this work, including at least one representative crystallographic model for each hit at room temperature and cryogenic condition.

Minor points:

Figure 3 panel A. Axis units are not readable. Please improve

L142 It would be interesting to know what is the operational temperature and humidity range of the sample chamber.

Line 325, Please include the organisms of origin of FosA

Reviewer #2

(Remarks to the Author)

Crystallographic fragment screening is an established method in drug development by now. Fragment libraries are screened for binding to a target protein and obtained hits can be developed into more potent binders. Due to automation and powerful synchrotrons entire fragment libraries can be screened in reasonable time. Experiments are usually performed at low temperature to minimize radiation damage. However, low temperature structures might not reflect physiological states. Here, the authors use serial crystallography at room temperature for fragment screening and compare this to conventional low temperature data collection from single crystals. This is an exhaustive study, showing possibilities and limitations of room temperature fragment screening.

Following are a few comments and suggestions:

line 156: "we collected several datasets from a subset of fragments.." Please specify how many datasets were collected.

line 182: "Interestingly, in both apo structures we observe residual difference density in this region, indicating high mobility." Is this mobility reflected in elevated B-values for the loop region? Please comment.

line 210: "This fragment was not seen to bind in the RT2 screen..." Even in the RT1 screen the presence of G08 appears not too convincing according to the density presented in Figure 5i. Also, according to the PDB validation report, the RSCC is only 0.74, which is on the low side for a ligand. However, this might not be the final value, as the PDB validation report is marked "not for manuscript review" and does not contain the PDB entry number.

line 226: "In the datasets RT1 and RT2 three and two fragments, respectively, are found at this site." According to Figure 4f, it is three fragments for RT1 (E07, E12, H01) and three for RT2 (E04, E07, H01). Please check also the numbers for the cryo screens. According to Figure 4f, there are 22 for cryo1 and 17 for cryo2.

line 211 and line 261: Differences in RT screens are described as "experimental uncertainties" and "uncontrolled parameters". It seems feasible, that there are different hit numbers in the cryo screens 1 and 2, due to different crystal size and thus resulting in better data quality for cryo1. However, the observed differences for the room temperature screens are not satisfactorily. Three out of seven fragments (43%) are only observed in RT1 or RT2. In large screening campaigns it is hardly feasible to perform screens multiple times. It would be nice if the authors could come up with a more plausible explanation, maybe go back to the lab journal to check for differences or discuss if some of the single events were borderline cases.

line 302: "The resulting structural information... will directly facilitate ligand optimization..." While the reviewer fully agrees with the statement, the here observed lower number of hits at room temperature limits the chemical synthesis options for medicinal chemists.

line 420: Was the fragment included in the cryo protection mixture?

References: In references 2, 6, 16, 24 the page numbers are missing.

line 692 (Figure 2): Overview of resolution of datasets is Figure 2b not a, and Rfree is Figure 2c not b.

For readers not familiar with PanDDA event and Z-maps it would be nice if the authors could provide conventional Fo-Fc maps in the supplemental material.

Supplementary Table 1:

The CC* value for RT2 is listed as the same for all data and the highest resolution shell, correct?
The number of Ligand/ion atoms is listed as 2 for RT1 and RT2 and 6 for cryo1, cryo2 and RT_single. Clearly, there are some fragments with more than 2 or 6 atoms. Please correct.
Is a CC* value of 0.000 as given for the highest resolution shell of RT_single reasonable?

Supplementary Table 2:

For the deposited structures it should be mentioned if the structures were derived from the first (RT1, cryo1) or second (RT2, cryo2) dataset in case the fragment was observed in both.

Do the authors have an explanation why the Wilson B-values in the Table deviate from the values given in the PDB-validation reports?

In case the fragment occupancy deviates from 1.0, the occupancy should be listed in the table.

Reviewer #3

(Remarks to the Author)

The abstract effectively captures the scope of the study and engages the reader. The introduction is well-structured, clearly presenting the motivation and context, and setting up the excellent experimental design that follows.

While the primary aim of the study is to investigate differences in fragment screening, the identification of an alternative loop conformation in a previously well-characterized target is particularly noteworthy. This finding underscores that knowledge of a static structure is not always sufficient and highlights the importance of intrinsic conformational dynamics—even under crystalline conditions—for understanding function and guiding drug development.

The experimental design is rigorous, and the fact that the authors repeated the screening experiment—despite the inherent complexity and scale—is commendable. This should serve as a benchmark for future method development work. Although the detailed differences in fragment binding between cryogenic and room temperature datasets are of limited personal interest, they are likely to be of significant relevance to researchers focusing on protein–ligand interactions. In particular, the inclusion of replicate datasets is valuable for computational and machine learning model development. Such data are increasingly essential for advancing the field.

I cannot fully assess the relevance of the less frequently identified fragments in the room-temperature (RT) study compared to the cryogenic one. However, if the authors' reasoning is correct—that RT crystallography preferentially captures more biologically relevant fragments—then, together with the relative ease of sample preparation (as noted by the authors and in my own experience), this could become highly interesting for the pharmaceutical industry, which currently relies almost exclusively on cryogenic crystallography.

The methodology is sound, the data analysis is thorough and appropriately cautious, and the conclusions are supported by the evidence presented. The manuscript meets the expected standards of the field, and sufficient methodological detail is provided for reproduction.

Overall, this is an exciting and well-executed study. I recommend publication, subject to addressing the minor points listed below.

Minor comments:

- I do not see clear evidence for Fragment G08 in the data provided via Figshare or in Figure 5, panel I. Are the authors certain this ligand is present? I recommend computing a polder map. The provided event map does not cover the model. I managed to visualize it using symmetry mates, but this workaround is suboptimal. Please re-calculate the event map. As it stands, the map may not support the ligand's presence convincingly.
- While the apo data are non-isomorphous, the fragment datasets are not. An isomorphous difference map between A09 and G08 shows clear positive density at A09 (attached screenshot), but no corresponding density at G08 (attached screenshot), where negative density would be expected after subtracting G08. This suggests G08 may not be real.
- Given the ease with which an isomorphous difference map reveals a potentially false positive, the authors should re-evaluate all fragment hits using isomorphous difference maps. Serial crystallography's inherent isomorphism is a major strength and should be fully leveraged here. We explored this concept in a preliminary way in [<https://www.nature.com/articles/s41467-023-43523-5>], but the authors have ideal data for a more comprehensive assessment. I strongly encourage including such an analysis—it would significantly strengthen the manuscript.
- Line 167: The observed effect likely reflects typical cryo-compression. Please consider citing this relevant article: <https://www.ncbi.nlm.nih.gov/pmc/articles/PMC6130464/>
- Line 248 onwards: The assumption presented here is an interesting discussion point and should be elaborated further. Also, there is a typo: "...by data collected AS cryogenic..." → should be "...at cryogenic..."

Version 1:

Reviewer comments:

Reviewer #1

(Remarks to the Author)

Referee2 stated that "Line 211 and line 261: Differences in RT screens are described as "experimental uncertainties" and "uncontrolled parameters". It seems feasible, that there are different hit numbers in the cryo screens 1 and 2, due to different crystal size and thus resulting in better data quality for cryo1. However, the observed differences for the room temperature screens are not satisfactorily. Three out of seven

fragments (43%) are only observed in RT1 or RT2. In large screening campaigns it is hardly feasible to perform screens multiple times. It would be nice if the authors could come up with a more plausible explanation, maybe go back to the lab journal to check for differences or discuss if some of the single events were borderline cases."

In their rebuttal letter possible causes for this variability are discussed, including experimental variation, borderline interpretation of hits and data processing, but in the revised text only the first cause is mentioned. Please, address this issue in the main including a discussion of all possible causes.

In the discussion section of the new MS the authors have introduced the phrase "..., presumably hinting at a current strong prediction bias of the algorithm towards cryogenic temperatures. This is likely caused by the training data from the PDB of which as of July 2025 about 94 % are derived from structures collected at temperatures below 220 K."

However, structures of the protein investigated here were available in the PDB as early as 2018, which implies they have, with all likelihood, been included in the training set of AlphaFold2/3. This is a more likely and direct source of bias. Please, include this in the discussion.

Reviewer #2

(Remarks to the Author)

The authors addressed all comments and suggestions made in the previous report. Therefore, I have no more objections to publish this manuscript. There are only two minor corrections suggested:

1. In Figures S5 and S6 the column headers of column 1 and 3 state initial and final 2Fo-Fc map, whereas actually both 2Fo-Fc and Fo-Fc are shown in the columns.
2. In the rebuttal letter the authors state: "To increase consistency, the WilsonB values as stated in the Table for average values from the automatic refinement of the serial crystallography datasets are now derived from phenix.refine instead of the data processing step using CrystFEL." However, some values in the Table seem to be unchanged, eg. 9G1A and 9G1B.

Reviewer #3

(Remarks to the Author)

My comments were fully addressed and I believe this version of the manuscript is ready to publish, especially since now the relevant datasets are all in the PDB the reader can analyse all of the fragment hits for themselves.

Response to reviewer's comments

We thank all reviewers for the time they took to review our manuscript and the comments they provided. We believe that in addressing these, we could improve our manuscript. Please find our detailed responses below.

REVIEWER COMMENTS

Reviewer #1 (Remarks to the Author):

The authors present a method for room temperature fragment screening using a serial X-ray crystallography approach. The unique aspect of this method is the use of a microporous solid-target sample chip whose design has been improved for this specific application and a specialized data collection chamber that keeps constant temperature and humidity. A library of 95 small molecule fragments is screened against the FosA protein from *Klebsiella pneumoniae* at room temperature and with standard cryogenic approaches, in replicate to enable systematic comparison. This led to the identification of 31 fragment hits binding in 8 distinct binding sites. Only 7 out of the 31 hits were identified at room temperature and for those common at both temperatures binding sites and binding poses were identical. All the hits identified at RT were also found under cryogenic conditions. However, an alternate conformation of the potassium binding loop was also found at RT. The authors attribute differences in ligand binding to increased thermal motion and disorder at room temperature, highlighting its potential relevance for drug design. The manuscript is well written and concise establishes proof of concept for the method and presents valuable results.

However, in this reviewer's view, a number of points need to be addressed:

1. The abstract, lines 33-35, states: "But overall. We observed more binders at cryo, however mainly at physiologically likely less relevant sites"

I find this phrase could be misinterpreted. It might be understood that cryogenic experiments might not have identified pharmacologically relevant or all pharmacological relevant sites in this experiment. But the results show that all fragment hits identified at RT were also identified at cryogenic temperature. Moreover, cryogenic experiments identified more fragment hits at pharmacologically relevant sites than the RT approach. Please, revise this phrase to clarify. For example: "We observed more binders at cryo, both at physiologically relevant and non-relevant sites"

We thank the reviewer for this suggestion and we have implemented the suggested changes to clarify this sentence. It now reads: "*But overall, we observed more binders at cryo, both at physiologically relevant and non-relevant sites.*" (lines 34f)

2. Figure 2 Panel B. RT1 and RT2 data has been grouped in bins of 0.1 angstroms. Instead of continuous values which gives a strange appearance to the graph. However, this makes comparison with the cryogenic data difficult and no explanation for this is given. Please plot continuous values for RT data as well.

The data was not grouped in 0.1 Å bins for the plot in the figure but the resolution cut-off of the serial crystallography data was selected by 0.1 Å steps for the downstream refinement and calculation of data processing statistics (as mentioned in the “SX data processing” paragraph in the Methods section). To enable a more balanced comparison of the achievable resolution limits of the datasets, we estimated the resolution at which $I/\sigma I=1$ by linear interpolation. This value is now used for Fig. 2b as well as Supplementary Fig. 3a and stated in Supplementary Table 1. The mean resolution for RT1 changed from 1.43 ± 0.08 Å to 1.45 ± 0.08 Å, and for RT2 from 1.48 ± 0.1 Å to 1.49 ± 0.08 Å. However, for the downstream analysis (automatic refinement, PanDDA analysis) the originally determined resolution limits were used.

The following text was added to the method section:

“To enable a better comparison of data processing statistics with the single crystal datasets, a more precise resolution limit was estimated by linear interpolation to determine the resolution at which $I/\sigma I=1$.” (lines 427f)

3. In the results section (line 130-145) a description of how the chips are prepared for data collection is introduced, but not of the previous steps. i.e. how crystals are grown and ligands delivered to the crystals, please include a succinct description of the whole protocol including these steps in the results section. This will facilitate understanding both by specialist and non-specialist readers.

The description in the manuscript now contains more details about the procedure. We have further added a new Supplementary Fig. 1 with pictures showing the sample preparation process from crystallization over fragment application to final cover sleeve addition before data collection. Additionally, we expanded the description in the results section:

“Crystals were directly grown in the compartments of the sample holders using on-chip crystallization and 3D-printed crystallization chambers (Supplementary Fig. 1a-d). This procedure is based on the commonly applied method of sitting-drop vapor-diffusion and adapted for fixed-target serial crystallography (27). After crystals grew to sufficient size, the crystallization solution was removed by blotting through the pores of the sample holder and solutions containing the fragments were added to the crystals in the compartment by pipetting (Supplementary Fig. 1e-i). After 24 h incubation time, excess liquid was again removed by blotting through the microporous membranes and a protective cover was slid over each of the sample holders (Supplementary Fig. 1j-m). (lines 135 ff)

4. Line 290. “our method avoids the most time-consuming steps required for conventional screening experiments. These include the search for suitable cryo-protectant, soaking individual crystals with ligands, loop-mounting individual crystals, flash-cooling, as well as mounting of the individual crystals on the goniometer and centering in the X-ray beam.”

This reviewer disagrees with this statement. In this method, soaking of crystals is still necessary (not single but multiple microcrystals). Moreover, protecting sieves have to be applied to the chips to avoid crystal dehydration once out of the crystallization chamber, all of which are probably manual operations. Similarly mounting of individual crystals in a goniometer is replaced by mounting of the chips into the temperature-controlled chamber and the necessary preparations for X-ray measurements. In cryogenic screening most of these operations are carried out with robotic equipment and automated sample exchangers that present hundreds of samples to the X-ray beam in fully automated mode making the process very efficient and probably more efficient than the process presented here. Adding to this, the authors do not compare the efficiency of their method with other RT approaches, like for example in situ data collection where samples are measured directly in crystallization plates avoiding crystal manipulations (refs. 8 and 15). While the RT screening method presented here has a value of its own, I disagree with the notion that it is less time consuming or more efficient than existing cryogenic approaches, which are highly automated, at least at this point. Indeed, while measuring collections of 500-1000 fragments is standard with cryogenic data collection, only 95 fragments were measured in this work, which are relatively low numbers in the context of fragment screening. A more balanced comparison of this approach, cryogenic methods and other RT methods previously applied to ligand screening should be presented in the discussion.

As suggested by the reviewer we have expanded our discussion accordingly and added the following text:

“While crystal growth on the sample holders, ligand application, blotting, cover sleeve application and mounting on the beamline were conducted manually in the present study, it has to be noted that each sample holder carries 12 different samples and blotting, sleeve application and mounting only need to be done once per sample holder, thereby increasing throughput. Moreover we are working on automation of these steps to further reduce manual intervention and increase reproducibility as well.

In situ data collection directly in crystallization plates is another approach for RT screening that circumvents the need for mounting individual crystals (8, 15). Typically, multiple crystals are merged together to obtain complete datasets. This can also help to reduce the detrimental effect of radiation damage that limits resolution from single crystals at RT and improve the overall resolution. A potential drawback compared to the serial data collection presented in this study is the increased scattering background originating from the crystallization plate and the crystallization droplet.

In particular at X-ray free electron lasers, high-viscosity extruders are used for sample delivery in serial crystallography experiments. Based on this method multi-sample devices have been developed (30), however at the moment these do not provide a high enough throughput for larger screening campaigns. Furthermore, the background from the sample medium could be a concern reducing data quality.” (lines 306ff)

5. Only 19 PDB depositions are mentioned linked to this work: 9G1A, 9G1B, 9G1C, 9G1D, 9G1E, 9G1F, 9G1G, 9G1H, 9G1I, 9G1J, 9G1K, 9G1L, 9G1M, 9G1N, 9G1O, 9G1P, 9G1Q, 9G1R, 9G1S. However, the authors report 31 fragment hits were found under cryogenic conditions for which 7 were also found at room temperature. Why not all models for fragment hits have been deposited?

One would expect 38 depositions associated to this work, including at least one representative crystallographic model for each hit at room temperature and cryogenic condition.

We have now also fully refined and deposited the models for the 21 hits found at the surface sites 5-7, which were only observed in the cryo datasets. Now a model for each hit at room-temperature and cryogenic temperature has been deposited.

Minor points:

Figure 3 panel A. Axis units are not readable. Please improve

We have increased the font size of the axis labels.

L142 It would be interesting to know what is the operational temperature and humidity range of the sample chamber.

The sample chamber can be reliably operated in the range of 7-40 C and 20-100% r.h. The information has been added to line 149:

“The setup includes a sample chamber which allows the precise control of temperature and r.h. during data collection (7-40 °C, 20-100% r.h.).” (lines 148 f)

Line 325, Please include the organisms of origin of FosA

We have now written out the protein name including originating organism:

“Fosfomycin resistance protein A from Klebsiella pneumoniae (FosAKP) was expressed and purified as described (29).” (line 338)

Reviewer #2 (Remarks to the Author):

Crystallographic fragment screening is an established method in drug development by now. Fragment libraries are screened for binding to a target protein and obtained hits can be developed into more potent binders. Due to automation and powerful synchrotrons entire fragment libraries can be screened in reasonable time. Experiments are usually performed at low temperature to minimize radiation damage. However, low temperature structures might not reflect physiological states. Here, the authors use serial crystallography at room temperature for fragment screening and compare this to conventional low temperature data collection from single crystals. This is an exhaustive study, showing possibilities and limitations of room temperature fragment screening.

Following are a few comments and suggestions:

line 156: “we collected several datasets from a subset of fragments..” Please specify how many datasets were collected.

We have added more specific information. The sentence now reads: “[...], we collected 29 datasets from eight fragments of the F2X-entry screen (RT_single), including reference apo datasets,[...]” (lines 162f)

line 182: “Interestingly, in both apo structures we observe residual difference density in this region, indicating high mobility.” Is this mobility reflected in elevated B-values for the loop region? Please comment.

Indeed, this region’s higher mobility is also reflected in increased B values. To illustrate this, we added new Supplementary Fig. 4 and added the following text:

“Interestingly, in both apo structures we observe residual difference density in this region (Fig. 3c) and also elevated B factors (Supplementary Fig. 4), indicating high mobility.” (lines 191f)

line 210: “This fragment was not seen to bind in the RT2 screen...” Even in the RT1 screen the presence of G08 appears not too convincing according to the density presented in Figure 5i. Also, according to the PDB validation report, the RSCC is only 0.74, which is on the low side for a ligand. However, this might not be the final value, as the PDB validation report is marked “not for manuscript review” and does not contain the PDB entry number.

We apologize for the accidental inclusion of the preliminary PDB validation report instead of the final one (PDB 9GIG). This has now been updated. The RSCC of the ligand is still 0.74. We agree that identification of this fragment as binder is probably a borderline case as already seen by the failure to identify this fragment in RT2. Nevertheless, we do believe this to be a true positive hit in the screen. Also in response to comments from Reviewer 3 we have calculated conventional isomorphous difference maps as well as maps from extrapolated structure factors, that are optimized to detect low occupancy states, to provide supporting data (see new Supplementary Fig. 5).

line 226: “In the datasets RT1 and RT2 three and two fragments, respectively, are found at this site.” According to Figure 4f, it is three fragments for RT1 (E07, E12, H01) and three for RT2 (E04, E07, H01). Please check also the numbers for the cryo screens. According to Figure 4f, there are 22 for cryo1 and 17 for cryo2.

The reviewer is correct. Additionally, there was a mistake in the preparation of this figure. The three fragments observed at this site in RT2 are E04, E07 and E12. Furthermore, during refinement of the remaining structures exclusively found in the cryo screens at this site, it became apparent that site 5 and site 6 are related by crystal symmetry and can be grouped together. Also fragment H06 was observed additionally at site 6 (previously site 7). Subsequently, there are in total 23 fragments found at this site in cryo1 and 17 in cryo2. The text and Fig. 4f have been updated:

“In each dataset RT1 and RT2 three fragments are found at this site. In strong contrast, in the screens cryo1 and cryo2, 23 and 17 fragments are identified at this site.” (lines 236ff)

line 211 and line 261: Differences in RT screens are described as “experimental uncertainties” and “uncontrolled parameters”. It seems feasible, that there are different hit numbers in the cryo screens 1 and 2, due to different crystal size and thus resulting in better data quality for cryo1. However, the observed differences for the room temperature screens are not satisfactory. Three out of seven fragments (43%) are only observed in RT1 or RT2. In large screening campaigns it is hardly feasible to perform screens multiple times. It would be nice if the authors could come up with a more plausible explanation, maybe go back to the lab journal to check for differences or discuss if some of the single events were borderline cases.

PanDDA enables identification of weakly bound ligands. Increasing the homogeneity of the data increases the success rate of identifying bound ligands as was demonstrated by preclustering of datasets by cluster4x (Ginn, <https://doi.org/10.1107/S2059798320012619>; Mehlman, et al., <https://doi.org/10.1016/j.str.2024.05.010>) and as was employed in this study. Certainly, some of the ligands will be borderline cases that will sometimes be detected, sometimes not, depending on the determined groundstate average map, the reference map in PanDDA that is used to detect outliers in the electron density that might represent a ligand bound with low occupancy.

We observe a more homogenous unit cell distribution for RT2. For RT1 35% of all datasets (41) belonged to the smaller unit cell cluster, while for RT2 only 10% (14) did, providing a better basis for identification of bound fragment by PanDDA.

Also different data quality of individual fragment datasets between the two screen can be a source of failed reproducibility. For fragment H01 we observed, for example, a clear difference in the quality of both datasets:

	Resolution	CC1/2	I/sigI	Rwork/Rfree	Wilson B	B-factor
RT1	1.4	0.9671	6.2	0.1472/0.1656	15.35	20.73
RT2	1.5	0.9494	4.7	0.1572/0.196	19.12	24.57

It is likely that the worse quality of the RT2 dataset contributes to the failure to detect fragment H01 in the structure.

Until now all steps in sample preparation (crystal growth on the sample holders, ligand application, blotting, cover sleeve application and mounting on the beamline) are conducted manually, which likely contributes to the heterogeneity of the data between RT1 and RT2. We are working on further automation of these steps to increase reproducibility of data quality and hit identification.

The following sentences have been added to the discussion:

“While crystal growth on the sample holders, ligand application, blotting, cover sleeve application and mounting on the beamline were conducted manually in the present study, it has to be noted that each sample holder carries 12 different samples and blotting, sleeve application and mounting only need to be done once per sample holder, thereby increasing throughput. Moreover we are working on automation of these steps to further reduce manual intervention and increase reproducibility as well.” (lines 306ff)

line 302: “The resulting structural information... will directly facility ligand optimization...” While the reviewer fully agrees with the statement, the here observed lower number of hits at room temperature limits the chemical synthesis options for medicinal chemists.

We agree with the reviewer that more fragment hits are generally more helpful as they extend the chemical space for ligand optimization. The cited sentence appears in the context of screening ten times larger libraries, which will certainly yield more hits offering more chemical synthesis options as obtained from the limited set identified with the relatively small F2X entry screen.

line 420: Was the fragment included in the cryo protection mixture?

No, the fragment was not included in the cryo protection mixture. To prevent any confusion, we now explicitly state this in the method section.

“After fragment soaking for at least 24 h, crystals were briefly soaked in 16% (w/v) PEG3350, 0.25 M MgCl₂, 0.2 M KBr, 0.1 M BisTris, pH 5.5, 5% (v/v) DMSO, 10.5% EG without fragment for cryo protection, [...]” (line 457)

References: In references 2, 6, 16, 24 the page numbers are missing.

The references have been updated.

line 692 (Figure 2): Overview of resolution of datasets is Figure 2b not a, and Rfree is Figure 2c not b.

The figure legend has been updated.

For readers not familiar with PanDDA event and Z-maps it would be nice if the authors could provide conventional Fo-Fc maps in the supplemental material.

PanDDA maps along with 2Fo-Fc and Fo-Fc maps from the automatic refinement pipeline (before addition of ligand model) and final maps from refined models including ligands were added as Supplementary Fig. 5 and 6.

Supplementary Table 1:

The CC* value for RT2 is listed as the same for all data and the highest resolution shell, correct?

This is not correct and was a mistake during preparation of the table. The values have been updated.

The number of Ligand/ion atoms is listed as 2 for RT1 and RT2 and 6 for cryo1, cryo2 and RT_single. Clearly, there are some fragments with more than 2 or 6 atoms. Please correct.

At this initial step no fragments had been modeled but only a ligand-free model was used for automatic refinement. The 2 atoms correspond to the two Mn²⁺ ions and for the single crystal datasets, there was an additional ethylene glycol (4 atoms) used in the input model. To prevent any confusion, we added the following sentence to the methods section:

"At this point no fragments were modeled yet." (lines 419f)

Is a CC* value of 0.000 as given for the highest resolution shell of RT_single reasonable?

We agree that this would not be reasonable. This is again a mistake in the preparation of the table. The correct values for CC* are 0.997 ± 0.004 (0.804 ± 0.066). Supplementary Table 1 has been updated.

Supplementary Table 2:

For the deposited structures it should be mentioned if the structures were derived from the first (RT1, cryo1) or second (RT2, cryo2) dataset in case the fragment was observed in both.

A row referring to the screen from which the dataset originated has been added to the table.

Do the authors have an explanation why the Wilson B-values in the Table deviate from the values given in the PDB-validation reports?

We report the Wilson B as stated in the header of the PDB file derived from the phenix refinement. The Wilson B from the PDB validation report was calculated with phenix.xtriage. Indeed recalculating it with a phenix.xtriage results in a slightly higher value.

To increase consistency, the WilsonB values as stated in the Table for average values from the automatic refinement of the serial crystallography datasets are now derived from phenix.refine instead of the data processing step using CrystFEL.

In case the fragment occupancy deviates from 1.0, the occupancy should be listed in the table.

The table has been updated with the refined occupancies of the ligands as well as the 1-BDC values from the PanDDA analysis, which were used as an initial estimate for the occupancy of the ligand for refinement (initial occupancy = $2 \cdot (1 - \text{BDC})$) (see also ref. 41).

We also added the following sentence to the method section:

"The initial occupancy of the fragments was set to $2 \cdot (1 - \text{BDC})$ and subsequently refined during model refinement (Table S2) (40)." (lines 425 f)

Reviewer #3 (Remarks to the Author):

The abstract effectively captures the scope of the study and engages the reader. The introduction is well-structured, clearly presenting the motivation and context, and setting up the excellent experimental design that follows.

While the primary aim of the study is to investigate differences in fragment screening, the identification of an alternative loop conformation in a previously well-characterized target is particularly noteworthy. This finding underscores that knowledge of a static structure is not always sufficient and highlights the importance of intrinsic conformational dynamics—even under crystalline conditions—for understanding function and guiding drug development.

The experimental design is rigorous, and the fact that the authors repeated the screening experiment—despite the inherent complexity and scale—is commendable. This should serve as a benchmark for future method development work. Although the detailed differences in fragment binding between cryogenic and room temperature datasets are of limited personal interest, they are likely to be of significant relevance to researchers focusing on protein–ligand interactions. In particular, the inclusion of replicate datasets is valuable for computational and machine learning model development. Such data are increasingly essential for advancing the field.

I cannot fully assess the relevance of the less frequently identified fragments in the room-temperature (RT) study compared to the cryogenic one. However, if the authors' reasoning is correct—that RT crystallography preferentially captures more biologically relevant fragments—then, together with the relative ease of sample preparation (as noted by the authors and in my own experience), this could become highly interesting for the pharmaceutical industry, which currently relies almost exclusively on cryogenic crystallography.

The methodology is sound, the data analysis is thorough and appropriately cautious, and the conclusions are supported by the evidence presented. The manuscript meets the expected standards of the field, and sufficient methodological detail is provided for reproduction.

Overall, this is an exciting and well-executed study. I recommend publication, subject to addressing the minor points listed below.

Minor comments:

- I do not see clear evidence for Fragment G08 in the data provided via Figshare or in Figure 5, panel I. Are the authors certain this ligand is present? I recommend computing a polder map. The provided event map does not cover the model. I managed to visualize it using symmetry mates, but this workaround is suboptimal. Please re-calculate the event map. As it stands, the map may not support the ligand's presence convincingly.

There is an issue of the *.ccp4 map files generated by the used PanDDA version (0.2.14) and representation in newer coot versions (expansion of the map to neighboring unit cells). In coot 0.8.9.1 the maps are displayed correctly, but not, for example, in 0.9.8.92. In Figshare we had uploaded the original maps from the PanDDA analysis. To circumvent this issue, we converted the ccp4 maps into mtz structure factor files and uploaded these together with all autorefined models to zenodo (<https://doi.org/10.5281/zenodo.15863149>).

We are certain that fragment G08 is present in the dataset based on the presented PanDDA event and Z-map. Please see also answer to comment below.

- While the apo data are non-isomorphous, the fragment datasets are not. An isomorphous difference map between A09 and G08 shows clear positive density at A09 (attached screenshot),

but no corresponding density at G08 (attached screenshot), where negative density would be expected after subtracting G08. This suggests G08 may not be real.

We agree with the reviewer that isomorphous difference (Fo-Fo) maps are a very sensitive tool to detect subtle differences between two datasets, in particular for serial crystallographic datasets. In our experience the PanDDA maps demonstrate improved sensitivity for detection of weakly bound ligands where a larger set of datasets is available. Fragment A09 is a covalent binder and displays full occupancy leading to a strong signal in an isomorphous difference map. In contrast fragment G08 binds non-covalently and more weakly and, hence, displays much lower occupancy (0.39) and we would also expect a smaller signal in an Fo-Fo map.

In the light of the reviewer's suggestion (see comment below) we explored isomorphous difference maps for confirmation of ligand detection for our set of RT fragment hits. As mentioned by the reviewer, the data are non-isomorphous, making PanDDA analysis more difficult. But the RT data can be broadly separated into two clusters based on the length of the unit cell's a-axis. The cluster with the larger unit cell (a-axis > 70.5 Å) contains the FosAKP apo structure. Therefore, this dataset was used as reference for Fo-Fo maps from hit datasets from this cluster (A12, E04, H01). Because we don't have a FosAKP apo structure from the cluster with the smaller unit cell (a-axis ≤ 70.5 Å), we selected a dataset from this cluster that presumably did not have a bound fragment. We sorted all datasets from this cluster based on their Rfree value and filtered out any datasets from fragments that had been identified in the cryo screens. This led to data D03 ("xg-new_FAKP_F2X_chipD_grid_fly_001_window_9") as reference dataset for the second cluster. We then calculated q-weighted Fo-Fo maps. In addition, we also calculated extrapolated structure factors and derived 2Fextrapolated-Fc maps to obtain a map with enhanced signal of the partially bound fragments. Both maps are shown in the new Supplementary Fig. 4. While the signal strength for the bound fragments is varying in the Fo-Fo maps, the extrapolated maps show clear signal for all fragments.

In addition to Supplementary Fig. 4 we added the following text to the method section:

"A common method to detect weak signals in serial crystallographic datasets, in particular in time-resolved studies, is the use of isomorphous difference maps (Fo-Fo maps). For the fragments identified by PanDDA we checked the signal for the fragments in the data by calculating q-weighted Fo-Fo maps (42). For datasets with an a axis > 70.5 Å we used the FAKP apo dataset as reference. Because there was no FosAKP apo dataset in the cluster with the smaller unit cell (a axis ≤ 70.5 Å), we selected a dataset from this cluster that presumably did not have a bound fragment. For this, we sorted all datasets from this cluster based on their Rfree value and filtered out any datasets from fragments that had been identified in the cryo screens. This led to D03 ("xg-new_FAKP_F2X_chipD_grid_fly_001_window_9") as reference dataset for the second cluster. We used Xtrapol8 to calculate q-weighted Fo-Fo maps (42). Additionally we calculated extrapolated structure factors from these and derived 2Fextrapolated-Fc maps that show enhanced signal also for weakly bound ligands (42)." (lines 431ff)

- Given the ease with which an isomorphous difference map reveals a potentially false positive, the authors should re-evaluate all fragment hits using isomorphous difference maps. Serial crystallography's inherent isomorphism is a major strength and should be fully leveraged here. We explored this concept in a preliminary way in [<https://www.nature.com/articles/s41467-023-43523-5>], but the authors have ideal data for a more comprehensive assessment. I strongly encourage including such an analysis—it would significantly strengthen the manuscript.

Please see answer to the comment above for a response

- Line 167: The observed effect likely reflects typical cryo-compression. Please consider citing this relevant article: <https://www.ncbi.nlm.nih.gov/pmc/articles/PMC6130464/>

We agree with the reviewer that the observed shrinkage of the unit cell at cryogenic temperature is likely caused by cryo compression, which is observed for the large majority of crystalline materials. Therefore we consider this as trivial observation. The reference above is a valuable analysis of the more specific effects of different cryo-solvents on thermal motions in crystals, but appears not relevant to this generic observation.

- Line 248 onwards: The assumption presented here is an interesting discussion point and should be elaborated further. Also, there is a typo: “...by data collected AS cryogenic...” → should be “...at cryogenic...”

We have expanded this sentence to include the bias towards cryogenic temperatures in the training data in the PDB. It now reads:

“[...] presumably hinting at a current strong prediction bias of the algorithm towards cryogenic temperatures. This is likely caused by the training data from the PDB of which as of July 2025 about 94 % are derived from structures collected at temperatures below 220 K. The new conformation of FosAKP observed at RT [...]” (lines 259 ff)

Further changes to the manuscript:

Besides the changes made in response to the reviewer’s comments, we corrected further issues that we recognized during preparation of this revision:

- Rwork values as reported in Supplementary Table 1 for refined structures were wrong. These were taken from the Rvalue (working + test set) instead of Rvalue (working set) in the header of the refined pdb files (as reported by phenix.refine). This has been corrected.
- During reevaluation of the cryo hits for this revision we realized that fragment binding site 5 is equivalent to site 6 due to the symmetry of the crystal. Therefore, we grouped all hits from site 5 and 6 together. Subsequently, there are only 7 binding sites altogether. The numbering in the text as well as Fig. 4 have been updated accordingly.
- Fig. 4f (overview of hits identified in all screens) was updated because H01 was mistakenly labeled as hit in RT2 while E12 was not. This has been corrected.
- Due to addition of Supplementary Fig. 1, 4, 5 and 6 the other figure numbers were updated.

Changes made to comply with Nature Communications policies and formatting guidelines:

- We added a table (Supplementary Data 1) with the source of the fragments used in the F2X entry library (vendor, order number, lot number) and added the following sentence to the method section:

“The fragments were acquired from various vendors (for details incl. order and lot numbers see Supplementary Data). No further characterization was conducted.” (lines 354ff)

- We changed Supplementary Figure label from “Fig. SX” to “Supplementary Fig. X”
- We moved the Acknowledgment section behind References.
- We added author contributions and competing interest statements.
- The data availability statement was updated.
- A code availability statement was added.
- Supplementary Notes were moved behind Supplementary Tables.

Response to reviewer's comments on revised manuscript

We thank the reviewers once more for their comments on the revised version of our manuscript. Please find our responses (in blue) to remaining issue below.

REVIEWERS' COMMENTS

Reviewer #1 (Remarks to the Author)

Referee2 stated that "Line 211 and line 261: Differences in RT screens are described as “experimental uncertainties” and

“uncontrolled parameters”. It seems feasible, that there are different hit numbers in the cryo screens 1 and 2, due to different crystal size and thus resulting in better data quality for cryo1. However, the observed differences for the room temperature screens are not satisfactorily. Three out of seven fragments (43%) are only observed in RT1 or RT2. In large screening campaigns it is hardly feasible to perform screens multiple times. It would be nice if the authors could come up with a more plausible explanation, maybe go back to the lab journal to check for differences or discuss if some of the single events were borderline cases."

In their rebuttal letter possible causes for this variability are discussed, including experimental variation, borderline interpretation of hits and data processing, but in the revised text only the first cause is mentioned. Please, address this issue in the main including a discussion of all possible causes.

We have now expanded the discussion and added the following text:

“The hit identification method PanDDA is depending on the homogeneity of the analyzed data, and preclustering of datasets, as was also done here, improves this (15, 30). Some of the identified ligands were likely borderline cases, the detection of which depends on the generated reference map in PanDDA. Therefore, subtle differences in diffraction quality and homogeneity of data between RT1 and RT2 could lead to a failure, for example, in identifying ligand H01 in both screens.” (p7, l280ff)

In the discussion section of the new MS the authors have introduced the phrase

"..., presumably hinting at a current strong prediction bias of the algorithm towards cryogenic temperatures. This is likely caused by the training data from the PDB of which as of July 2025 about 94 % are derived from structures collected at temperatures below 220 K."

However, structures of the protein investigated here were available in the PDB as early as 2018, which implies they have, with all likelihood, been included in the training set of AlphaFold2/3. This is a more likely and direct source of bias. Please, include this in the discussion.

We thank the reviewer for pointing this out. We have added the following sentence to the discussion:

“Moreover, cryo structures of FosAKP were likely included in the training dataset.” (p7, l267f)

Reviewer #2 (Remarks to the Author)

The authors addressed all comments and suggestions made in the previous report. Therefore, I have no more objections to publish this manuscript. There are only two minor corrections suggested:

1. In Figures S5 and S6 the column headers of column 1 and 3 state initial and final 2Fo-Fc map, whereas actually both 2Fo-Fc and Fo-Fc are shown in the columns.

Thank you for observing this. We have adjusted the column headers of Supplementary Fig. 5 and 6 accordingly.

2. In the rebuttal letter the authors state: "To increase consistency, the Wilson B values as stated in the Table for average values from the automatic refinement of the serial crystallography datasets are now derived from phenix.refine instead of the data processing step using CrystFEL." However, some values in the Table seem to be unchanged, eg. 9G1A and 9G1B.

There may be a confusion. The Wilson B values for the individually refined structures as reported in Supplementary Table 2 have already in the previous manuscript version been those derived from phenix.refine. Therefore, these are unchanged. In contrast, the average values from the automatically refined structures from screens RT1 and RT2 reported in Supplementary Table 1 had previously been retrieved from the data processing in CrystFEL. For consistency the average Wilson B values for screens RT1 and RT2 are now also reported from phenix.refine. Therefore, only the value in Supplementary Table 1 have been updated.

Reviewer #3 (Remarks to the Author)

My comments were fully addressed and I believe this version of the manuscript is ready to publish, especially since now the relevant datasets are all in the PDB the reader can analyse all of the fragment hits for themselves.